# Sodium-Glucose Cotransporter-2 Inhibitors in Patients with Hereditary Podocytopathies, Alport Syndrome, and FSGS: A Case Series to Better Plan a Large-Scale Study

**DOI:** 10.3390/cells10071815

**Published:** 2021-07-18

**Authors:** Jan Boeckhaus, Oliver Gross

**Affiliations:** Clinic for Nephrology and Rheumatology, University Medical Center Göttingen, 37075 Göttingen, Germany; jan.boeckhaus@med.uni-goettingen.de

**Keywords:** podocytopathies, hereditary kidney diseases, Alport syndrome, focal segmental glomerulosclerosis, kidney therapies, nephroprotection, sodium-glucose cotransporter-2 inhibitors

## Abstract

Hereditary diseases of the glomerular filtration barrier are characterized by a more vulnerable glomerular basement membrane and dysfunctional podocytes. Recent clinical trials have demonstrated the nephroprotective effect of sodium-glucose cotransporter-2 inhibitors (SGLT2i) in chronic kidney disease (CKD). SGLT2-mediated afferent arteriole vasoconstriction is hypothesized to correct the hemodynamic overload of the glomerular filtration barrier in hereditary podocytopathies. To test this hypothesis, we report data in a case series of patients with Alport syndrome and focal segmental glomerulosclerosis (FSGS) with respect of the early effect of SGLT2i on the kidney function. Mean duration of treatment was 4.5 (±2.9) months. Mean serum creatinine before and after SGLT-2i initiation was 1.46 (±0.42) and 1.58 (±0.55) mg/dL, respectively, with a median estimated glomerular filtration rate of 64 (±27) before and 64 (±32) mL/min/1.73 m^2^ after initiation of SGLT2i. Mean urinary albumin-creatinine ratio in mg/g creatinine before SGLT-2i initiation was 1827 (±1560) and decreased by almost 40% to 1127 (±854) after SGLT2i initiation. To our knowledge, this is the first case series on the effect and safety of SGLT2i in patients with hereditary podocytopathies. Specific large-scale trials in podocytopathies are needed to confirm our findings in this population with a tremendous unmet medical need for more effective, early on, and safe nephroprotective therapies.

## 1. Introduction

Hereditary diseases of the glomerular filtration barrier are characterized by a more vulnerable glomerular basement membrane (GBM) and dysfunctional podocytes, which are unable to permanently tolerate hyperfiltration and elevated capillary pressure [1]. The type IV collagen disease Alport syndrome (AS) is the most common hereditary podocytopathy, characterized by a weakened GBM and progressive renal fibrosis due to hemodynamic overload. AS can mimic focal segmental glomerulosclerosis (FSGS) and accounts for almost 3% of cases of chronic kidney disease (CKD) [2]. Variants in the slit-diaphragm genes NPHS1 and NPHS2, as well as INF2, are other classical hereditary causes of FSGS in adolescents and young adults [2,3]. In these hereditary diseases of the glomerular filtration barrier, angiotensin-converting-enzyme inhibitors (ACEis) have evolved as the cornerstone of treatment, starting pre-emptively in oligo-symptomatic toddlers with isolated micro-hematuria [4,5]. ACEi-mediated efferent arteriole vasodilatation plus the inhibition of the podocyte’s angiotensin system account for the profound nephroprotective effect in children with AS [2,4,5]. The earlier the start of ACEi, the better the protective effect on the vulnerable GBM and on the dysfunctional podocytes, with the potential to delay end-stage renal failure (ESRF) by decades. However, despite blockade of the renin-angiotensin-aldosterone system (RAAS), most hereditary diseases of the glomerular filtration barrier (such as hemizygous males with X-linked AS) still progress with a high risk for ESRF, raising the urgent need for add-on therapies.

Recent randomized clinical trials (RCTs) have demonstrated the nephroprotective effect of sodium-glucose cotransporter-2 inhibitors (SGLT2i) in CKD [6]. SGLT2-mediated afferent arteriole vasoconstriction via a tubuloglomerular feedback, additive to efferent arteriole vasodilatation by ACEis, is thought to be responsible for this robust nephroprotective effect [7]. The effect of SGLT2i is not limited to specific CKDs [6]. However, SGLT2-mediated correction of the hemodynamic overload of the glomerular filtration barrier is hypothesized to also be promising in hereditary diseases with a more vulnerable GBM and dysfunctional podocytes, such as AS and FSGS [7].

In the present study, we report data on the use of SGLT2i in a case series of patients with AS and FSGS. The goal of our investigation was to characterize the early effect on the kidney function and to assess the safety of initiation of therapy [7,8]. To our knowledge, this is the first case series on the effect of SGLT2i in patients with hereditary causes of CKD.

## 2. Materials and Methods

This observational, non-interventional prospective case series was started in six patients with the diagnosis of AS or FSGS. Diagnosis was confirmed by kidney biopsy, genetic testing, or both. All patients originated from the same outpatient clinic for hereditary kidney diseases at the University Medical Center Goettingen; all patients have been cared for in this outpatient clinic for many years and are seen in 3–6-month intervals. Patients consented to off-label treatment with once-daily 10 mg empagliflozin (*n* = 3) or dapagliflozin (*n* = 3). All patients provided written consent to evaluate their scientific data. For patients with AS, the registry and data storage, in conformity with GCP guidelines, were approved by the Ethics Committee of the University Medical Center Göttingen (AZ 10/11/06; renewed version in 2014 and 2020; ClinicalTrials.gov Identifier NCT 02378805). Albuminuria is reported as urinary albumin-creatinine ratio in mg/g creatinine. Kidney function is presented as eGFR (using the CKD-EPI formula) at baseline and at follow-up.

## 3. Results

All male patients received SGLT2i additive to RAAS-blockade. In the female patient, ACEi was stopped prior to the start of SGLT2i because of angioedema. In addition to ACEi and SGLT2i, one patient also received tacrolimus because of inflammatory bowel disease. Patient characteristics are presented in Table 1.

In brief, the two young patients with hereditary FSGS showed a good to very good response in respect to lowering albuminuria. However, the development of eGFR varied from +30 mL/min/1.73 m^2^ in patient 1, with the most benefit in reduction of proteinuria, to −17 mL/min/1.73 m^2^ in patient 2, with a NPHS2 plus INF2-variant as underlying disease. This resulted in an improvement of CKD-stage G2A3 to G1A3 in patient 1 and a change from G3aA3 to G3bA2 in patient 2 (Table 1). In patient 1, the improvement of eGFR after 11 months is most likely not only related to the start of SGLT2i, but also resulted from recovery of nephrotic syndrome with remission of hypoalbuminemia and fluid overload, therefore improving plasma-protein binding of medication, malnutrition, and other consecutive factors.

The four older patients with X-linked Alport syndrome showed a more uniform, consistent response to SGLT2-inhibition: in the female patient 3, SGLT2i was able to compensate for the withdrawal of the ACEi (because of angioedema), and the CKD level remained unchanged at G1A3. Senior Alport syndrome patients, patient 4 and 5, each with a quite stable kidney disease, both benefited from SGLTi with a reduction of their proteinuria without a profound reduction of their eGFR. The CKD level remained unchanged in patient 4 (G3bA3) and dropped from G3aA3 to G3bA2 in patient 5. Patient 6, with the shortest follow-up, did not show a short-term benefit in his proteinuria and dropped from CKD level G3aA3 to G3bA3.

When analyzing all six patients together, the mean age (standard deviation) was 40  (±17) years, the mean duration of treatment was 4.5  (±2.9) months, and follow-up was 3 to 11 months. Mean serum creatinine before and after SGLT-2i initiation was 1.46 (±0.42) and 1.58 (±0.55) mg/dL, with a median estimated glomerular filtration rate (eGFR; using the CKD-EPI formula) of 64 (±27) before and 64 (±32) mL/min/1.73 m^2^ after initiation of SGLT2i (Figure 1A). Mean urinary albumin-creatinine ratio in mg/g creatinine before SGLT-2i initiation was 1827 (±1560) and decreased by almost 40% to 1127 (±854) after SGLT2i initiation (Figure 1B). As described in the literature, eGFR dipped after the initiation of SGLT2i in most patients [8]. Overall, treatment was well tolerated; however, eGFR initially decreased by more than 30% in patient 2 with a NPHS2 plus INF2-variant as underlying cause of FSGS; as a consequence, blood pressure medication with a calcium antagonist was reduced.

## 4. Discussion

As a working hypothesis for our case series, we postulate that SGLT2-mediated correction of the hemodynamic overload of the glomerular filtration barrier represents a new and highly promising therapeutic approach in (hereditary) podocytopathies. In diseases such as AS and FSGS, a more vulnerable GBM and dysfunctional podocytes are unable to permanently tolerate hyperfiltration and elevated capillary pressure [1,7].

As a notable limitation, our case series only intends to serve as a proof of concept for better planning of large prospective interventional trials. Early genetic diagnosis in minors with AS or hereditary FSGS opens a window of opportunity for early therapy with SGLT2is. However, only a randomized placebo-controlled trial specific for this population will provide the high evidence level needed to justify treatment recommendations in children and young adults. For this reason, our case series addressed an initial safety concern that in hereditary diseases of the GBM, the tubular feedback loop to the macula densa could malfunction in a way that SGLT2i would result in severe acute renal failure. A similar approach has been used by our group to better plan and optimize power calculation in the EARLY PRO-TECT Alport trial [5], which resulted in a new international treatment recommendation for children two years and older with very early stages of AS [9].

In addition, the initial eGFR drop in one of our patients raises some safety concerns in young patients with suspected hyperfiltrating glomeruli, which underline the need for a RCT with SGLT2i specific for this more vulnerable population with hereditary diseases of the glomerular filtration barrier. Very promising, in patient 1, with the longest follow-up period of 11 months, eGFR already returned to baseline.

All six patients are still under treatment with SGLT2i. We did not observe acute renal failure or hypovolemia. Additionally, no bacterial or mycotic infections of the urinary tract were diagnosed. None of the patients had to stop SGLT2i therapy because of side effects.

In conclusion, therapy with SGLT2i on top of RAAS-blockade was well tolerated and effective in terms of the initial drop in eGFR and the lowering of albuminuria. As a proof of concept, SGLT2-mediated correction of the hemodynamic overload of the glomerular filtration barrier looks to be a very promising therapeutic approach in hereditary podocytopathies. Specific RCTs are needed to confirm our findings in this genetically well-defined population with a tremendous unmet medical need for more effective, early on, and safe nephroprotective therapies. SGLT2is have a profound potential to correct the hemodynamic overload additive to RAAS-blockade in these patients with the inspiring potential to further delay ESRF by years.

## Figures and Tables

**Figure 1 cells-10-01815-f001:**
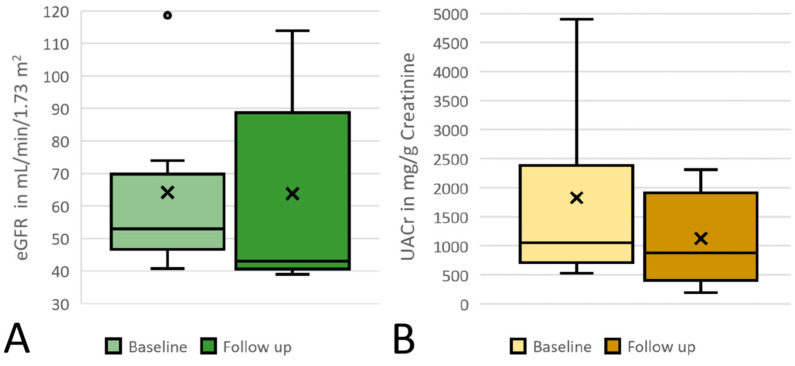
Proteinuria and kidney function; (**A**) eGFR at baseline and at follow-up; (**B**) UACr at baseline and at follow-up. x indicates the mean value; the circle indicates an outlier; eGFR: estimated glomerular filtration rate; UACr: urine-albumin-creatinine ratio; CKD: chronic kidney disease. × = mean; circle = outlier.

**Table 1 cells-10-01815-t001:** Clinical characteristics of patients.

					Baseline				Follow Up				
Patient Number	Age	Sex	Diagnosis	Co-Morbidity	CKD Stage	Crea mg/dL	eGFR	Albumin-Uria	CKD Stage	Crea mg/dL	eGFR	Albumin-Uria	Follow Up Time (months)
**1**	25	m	FSGS	IBD, aHT	G2A3	1.33	74	4900	G1A3	1.02	104	805	11
**2**	23	m	FSGS	sHP	G3aA3	1.66	57	530	G3bA2	2.24	40	192	3
**3**	35	f	XLAS	-	G1A3	0.61	119	706	G1A3	0.69	114	946	3
**4**	63	m	XLAS/FSGS	aHT, hearing loss	G3bA3	1.74	41	2719	G3bA3	1.87	39	2233	3
**5**	65	m	XLAS	aHT, sHP, hearing loss	G3aA3	1.49	49	713	G3bA2	1.64	43	273	3
**6**	30	m	XLAS/FSGS	aHT	G3aA3	1.92	46	1391	G3bA3	2.04	43	2314	4

m: male; f: female; FSGS: focal segmental glomerulosclerosis; XLAS: X-linked Alport syndrome; IBD: inflammatory bowel disease; aHT: arterial hypertension; sHP: secondary hyperparathyroidism; eGFR: estimated glomerular filtration rate; Crea: serum-creatinine.

## Data Availability

The full data set can be provided by the authors upon request.

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
