# Peer review of "Sodium-Glucose Cotransporter-2 Inhibitors in Patients with Hereditary Podocytopathies, Alport Syndrome, and FSGS: A Case Series to Better Plan a Large-Scale Study"

_cells, 2021, doi:10.3390/cells10071815_

Round 1
Reviewer 1 Report
Line
13 should say «with respect to the early effect of…..”
43 “most hereditary disease of the GFB…..still progress with a 100% lifetime risk” please provide reference or reformulate. Many pts with Alport’s (according to the definition by Kashtan et la 2018) do not necessarily progress to ESRF.
50 “ the effect of SGLT2i is not limited to specific CKDs” (please add reference of Dapa CKD trial)
52: please explain why SGLT2 I are “most promising” in hereditary diseases compared to diabetes or other glomerular diseases
54: please formulate exactly which hypothesis is tested. Or omit the sentence and stay descriptive.
61 I propose “AS and hereditary FSGS”
65 For what other reasons aside from proteinuria consented patients to SGLT2i treatment?
70: it should say “Albuminuria is reported as urinary albumin…….”
76: I assume the authors mean inflammatory bowel disease instead of irritable bowel disease?
78 why hereditary in brackets? Is it hereditary or not?
80: a GFR improvement of + 30 ml/min as a consequence of SGLT2i treatment does not make sense. Which mechanism is assumed to be accountable here? Please discuss
84 showed double
Table 1: IBD: inflammatory bowel disease (IBD). Please make a column with Patent 1, 2, 3 etc. Please state interval between baseline and follow up for each patient.
107 : calciumantagonist
Figure 1: what does the x stand for: mean?
Discussion:
Please state more clearly:What was your major goal in this analysis: to assess the amount of reduction of proteinuria? To investigate the effects of SGLT2 on eGFR. Other parameters? What was required for better planning of large prospective interventional trial?
126: see above: there must be another explanation for improvement of eGFR
134 why hereditary in brackets?
138 on top of and additive is redundant
139 “to delay ESRF by decades if not for their lifetime in most cases” very speculative and somehow exaggerated. Please rephrase more modestly
Reviewer 2 Report
This is a very interesting series of cases and I think that they will indeed be useful for helping to design trials that can provide more definitive answers on these potentially useful drugs. My main critique is that the authors could provide a lot more information about how these patients reacted to these medications. For example, they noted that the SGLT2 inhibitors can evoke an initial dip in eGFR but only provided values in a table and bar graph at the beginning and end of the observation period. I found myself wanting to see a time course assuming that they measured serum creatinine at multiple points during the period (as was implied in the text). For example, in one patient there was a notable decline in eGFR over the course of the treatment, even though that patient's albuminuria improved. I would have really liked to see what was going on with that patient in more detail. Was the decline in eGFR still part of a "dip" that was gradually improving? I was also hoping that the authors might have measured blood glucose and blood pressures in these patients at various time points, since it might be relevant to the design of future trials. In short, my main critique of this series of cases is not the design or the interest in the observations, it is that very little detail was provided.
